# Global Transcriptomic Profile Analysis of Genes Involved in Lignin Biosynthesis and Accumulation Induced by Boron Deficiency in Poplar Roots

**DOI:** 10.3390/biom9040156

**Published:** 2019-04-19

**Authors:** Wan-Long Su, Na Liu, Li Mei, Jie Luo, Yi-Jie Zhu, Zhu Liang

**Affiliations:** 1College of Horticulture and Forestry Sciences/Hubei Engineering Technology Research Center for Forestry Information, Huazhong Agricultural University, Wuhan 430070, China; wanlongsu@163.com (W.-L.S.); LNmiyi@163.com (N.L.); luojie@mail.hzau.edu.cn (J.L.); zyj2895@163.com (Y.-J.Z.); liang2915@foxmail.com (Z.L.); 2College of Biological Sciences and Technology, Beijing Forestry University, Beijing 100000, China

**Keywords:** boron deficiency, lignin biosynthesis, transcriptome, phytohormone, poplar

## Abstract

To uncover the transcriptomic mechanism of lignin accumulation caused by boron deficiency (BD), Nanlin895 (*Populus* × *euramericana* “Nanlin895”) was subjected to control (CK, 0.25 mg·L^−1^) and BD (0 mg·L^−1^) treatments for 3 days. RNA-Seq was carried out to survey the expression patterns of the lignin-regulated biosynthetic genes in response to BD. The results showed that 5946 genes were identified as differentially expressed genes (DEGs), 2968 (44.2%) of which were upregulated and 3318 (55.8%) of which were downregulated in response to BD. Among them, the expression of lignin monomer biosynthetic (*PAL*, *CCR*, *CAD*, *COMT*, *F5H*, *PER*/*LAC*) and modulated genes, for example, transcription factors (*MYBs*) and hormone signal regulating genes (*GIDs*, histidine kinase 1, coronatine-insensitive protein 1), were upregulated, and some hormone signal regulating genes, such as *AUXs* and BR-related (sterol methyltransferases), were downregulated under BD treatment. There are also some genes that were screened as candidates for an association with wood formation, which will be used for the further analysis of the function of lignin formation. These results provide an important theoretical basis and reference data in plant for further research on the mechanism of lignin accumulation under BD.

## 1. Introduction

Boron is one of the essential elements required for normal growth in higher plants [1]. It plays important roles in the physiological processes of cell wall structure and function, phenol metabolism, the structure and function of membranes, sugar transportation, carbohydrates, and nucleic acid metabolism [2,3,4]. Studies have shown that boron and the pectin polysaccharide RGII combine to form a dipolymer, which is involved in cell wall formation [5,6]. Boron deficiency (BD) results in the accumulation of pectinolytic-related substances and lignin in mature cells, inhibiting the formation and development of new tissues.

BD is a common phenomenon on a global scale, which is mainly caused by the leaching of rains in the soil [7]. Sillanpaa et al. [8] studied 190 samples of agricultural soil, and the results indicated that BD led to a 31% decrease in the yield of crops. With the emphasis on the harm and damage caused by BD, many studies have focused on crops, such as Arabidopsis [2], wheat [9], maize [10] and citrus [11,12]. From these studies, some genes, such as *EXP14*, *EXPB1*, *XTHs*, *CSLB5*, *PG*, and *PME*, related to the synthesis of the cell wall have been discovered under BD conditions [3]. However, until now, few researchers have focused on the cell wall synthesis of poplars under BD.

BD also leads to a decrease in forestry productivity. *Populus* is one of the most widespread trees that is planted all over the world because of its characteristics of fast growth, high yield, and strong adaptability [13,14]. It is a kind of raw material for the paper and pulp industries [15]. Lignin, a kind of phenol polymer, is an important component of the cell wall in vascular plants, and the content of lignin is an essential factor that influences the properties of the production of paper with poplars. In recent years, many researchers have made substantial efforts to breed planting clones with a lower lignin content; for example, some genetically modified poplars with a lower lignin content were obtained [16,17]. Thus, studies on the molecular mechanism of lignin accumulation after abiotic stresses (such as BD) are needed to breed low lignin poplar.

Previous studies by our group revealed that BD increased the lignin in citrus vessel cell walls [18]. In this paper, we propose that BD will cause the accumulation of lignin in poplar. To understand the molecular mechanism of lignin accumulation in poplars, poplar saplings were hydroponically cultured in nutrient solution with BD. Digital gene expression was performed to dissect the gene reprogramming profiles underlying lignin accumulation under BD.

## 2. Materials and Methods

### 2.1. Plant Materials and Treatment

Nanlin 895 (*P.* × *euramericana* “Nanlin895”) used in this study were from the poplar germplasm nursery of Shishou City in Hubei Province, China. In early July, two-month-old culture saplings approximately 30–40 cm in height with no pests, good growth, and uniform size were taken out from the pots, cleaned up, and precultured in nutrient solution (1/2 Hoagland’s nutrient solution and iron minor elements) for two weeks and cultured with deionized water for three days before BD treatment. During the treatment, the plants were divided into two groups randomly; a total of 168 poplar saplings with 12 independent biological replicates for each stage were included in each group and treated for different times (0, 1, 3, 5, 7, 10, and 15 days). Saplings were cultured using Hoagland’s nutrient solution containing either 0.25 mg·L^−1^ boron (control, CK) or 0 mg·L^−1^ boron (boron deficiency, BD). Poplar saplings were cultured in black plastic buckets ventilated for 15 min every two hours, and the nutrient solutions were replaced weekly. Root tips were collected to determine the content of lignin at different time points (0, 1, 3, 5, 7, 10, and 15 days) with twelve biological duplications. For the saplings, we used three plants to determine the lignin content, three for RNA-Seq, three for qRT-PCR, and others as additional materials. Then, the root tips with the significantly higher lignin content were selected to extract RNA using random selection method. All the nutrient solutions used in this experiment were prepared with deionized water.

### 2.2. Determination of Lignin Content

For lignin content determination, the fresh samples were heated for 20 min at 105 ℃, then heated at 65 ℃, and maintained at a constant weight. Then, the dried samples were ground into powder. Two methods (Klason and ultraviolet spectrophotometry (UV-6100, Mapada, Shanghai, China) were used to determine the lignin content. For the calculation of standard light absorption (A_standard_), Klason methods [19] were used to estimate the lignin content of the untreated samples. Absorptivity was calculated asAbsorptivity = Abs × liters/W_samples_ × k_lignin_
Astandard were calculated as average number of different Absorptivity of biological replicates.

For the lignin content of different stages of root samples, ultraviolet spectrophotometry was carried out [20]. Briefly, ca. 6 mg of root powder samples were mixed with 5 mL of 25% (w/w) acetyl bromide acetic acid and 0.2 mL of perchloric acid, sequentially. The tubes were sealed and incubated at 70 ℃ for 30 min, and then the mixture was mixed with 10 mL of 2 M NaOH and diluted with glacial acetic acid to 100 mL. The lignin contents in the samples were determined spectrophotometrically at 254 nm (the maximum absorption wavelength was obtained by scanning the absorption peak using ultraviolet-visible spectroscopy in the range of 200–600 nm). The calculation of the lignin contents was performed via the formula lignin% = Abs × liters × 100%/W_samples_ × A_standard_.

### 2.3. Total RNA Extraction and Digital Gene Expression Profiles

Root tips (ca. 3.5 cm from the top) at the stage of day 3 were collected for the following RNA isolation. Total RNA was extracted from the root tips of CK and BD treatments with 3 biological replicates at the same phase. RNA extraction was performed according to the manufacturer’s instructions of an RNA Plant Reagent (TIANGEN, Beijing, China).

RNA samples extracted from the root tips were sent to Shenzhen Huada Genomics Co (Huada, Shenzheng, China), Ltd. for quality and quantity evaluation, RNA-Seq library construction, and high throughput sequencing. In brief, RNA-Seq library construction was performed according to the operating instructions (MGIEasy RNA Library Prep Set, version 3.0 (Huada, Shenzheng, China). In brief, mRNA was enriched by using Oligo(dT) magnetic beads (Dynabeads® mRNA Purification Kit; Invitrogen, Cat. No. 61006, Carlsbad, CA, USA) from total RNA under high-temperature conditions. Then, the mRNA was broken into short fragments by the addition of fragmentation buffer. Then, the short fragments of mRNA were used as templates for the synthesis of first strand cDNA and second strand cDNA. The double string cDNA purified with magnetic beads was subjected to repair of the 3’ end, the addition of a dNTP base, and the ligation of sequencing adapters. The fragments were enriched using PCR amplification, and the products were purified using agarose gel electrophoresis. The quantity and quality of the completed cDNA libraries were tested using an Agilent 2100 Bioanalyzer (Agilent, Santa Clara, CA, USA) and an ABI Step One Plus Real-Time PCR System (Thermo Fisher Scientific, Santa Clara, CA, USA). The qualified library was sequenced with Illumina HiSeq TM2000 (Illumina, San Diego, CA, USA).

The clean reads were mapped to reference sequences of *P. trichocarpa* (ftp://ftp.jgi-psf.org/pub/compgen/phytozome/v9.0/Ptrichocarpa/annotation) SOAPaligner/SOAP2. Sequence saturation was used to analyze the number of genes detected that tended to saturate. The distribution of reads on the reference genes was used to evaluate the randomness.

### 2.4. Identification of Differentially Expressed Genes (DEGs)

The gene expression level was calculated by using the RPKM method (Reads Per kb per Million reads), as suggested by Wagner et al. [20].

The RPKM method can eliminate the influence of different gene sequencing discrepancies on the calculation of gene expression levels. Therefore, the RPKM values can be directly used for comparing the differences in gene expression among samples. The absolute value of log_2_ (BD-RPKM/CK-RPKM) was used to represent the fold change of two genes with a threshold FDR ≤ 0.001, and the absolute value of |log_2_Ratio| ≥ 1 was used to judge the significance of the difference in gene expression.

For the analysis of function and pathway, all the DEGs were mapped to GO terms in the database (http://www.geneontology.org/) and pathway terms in the KEGG database (http://www.genome.jp/kegg/), as described by Ye et al. [21] and Minoru et al. [22], respectively. GO term enrichment analysis was conducted in the agriGO database with the singular enrichment analysis (SEA) tool as described by Du et al. [23,24]. For functional categories, DEGs were submitted to MapMan software (version 3.6.0RC1, Golm, Germany) according to the standard protocol with minor modifications [24].

### 2.5. Quantitative Real-Time PCR Analysis

qRT-PCR analysis was used to verify the DEG results. Nine differentially expressed candidate genes involved in the biosynthetic pathway of lignin monomers and regulatory pathways were chosen for quantitative real-time PCR. For this analysis, gene-specific primers were designed according to the reference gene sequences using Primer 5.0 (Canada), and qRT-PCR was performed according to the ZHUANGMEN manufacturer specifications (Product code: Catalog # ZF101-102, Beijing, China). Briefly, a 20 µL reaction mixture included 10 µL 2× SYBR qPCR Mix, 0.4 µL Dye ROX, 0.5 µL cDNA temple, 0.4 µL forward primer (10 µM), 0.4 µL reverse primer (10 µM), and ddH_2_O to a final volume, and the reaction was performed on the 7500 Fast ABI Real-time PCR system (Applied Biosystems, Foster City, CA, USA) using a two-step method: 94 °C for 3 min, 94 °C for 15 s, and 60 °C for 1 min. The Poplar *actin2* gene was used as an internal reference [25], and the relative expression levels of the genes were presented by 2^-ΔΔCT^. The experiment was conducted with three biological reduplicates and three technological repeats. The primers used in this study are displayed in Appendix A.

### 2.6. Statistical Analysis

Statistical analysis was carried out with SPSS 17.0 (SPSS, Chicago, IL, USA). Data were tested for normal distribution, and one-way analysis of variance (ANOVA) was used to compare the differences between means. *p*-values lower than 0.05 were considered significant.

## 3. Results

### 3.1. Effect of BD on the Lignin Content in Poplar Roots

The root lignin content of different stages of “Nanlin895” under treatment and nontreatment conditions were determined using ultraviolet spectroscopy at a 254 nm wavelength. The lignin content gradually increased in the early stage of the treatment, peaked on day 3, and tended to remain constant for the following days (Figure 1). For the control, the lignin content was the lowest (28.42%) on day 3 and the highest (32.75%) on day 15 (Figure 1). In contrast, it was the highest on day 3 (39.28%), and the average content was remarkably higher than that of the control under boron deficiency conditions (Figure 1).

### 3.2. Read Mapping and Analysis of DEGs

To investigate the changes in gene expression in transcription in poplars under boron deficiency, poplar saplings were cultured in Hoagland’s nutrient solution containing either 0.25 mg·L^−1^ boron (control, CK) or 0 mg·L^−1^ boron (boron deficiency, BD). Root tips with the highest contents of lignin were used to extract total RNA and were further utilized to construct libraries (CK and BD); high-throughput sequencing was performed using Illumina HiSeq TM2000. More than 11 million clean reads, above 93% of the total raw reads, were obtained from each library (Appendix A). More than 7 million clean reads from each library were mapped to reference genes, and approximately 78% of the clean reads were mapped to the reference genome (Appendix A). Between BD and CK, 5944 genes had an FDR ≤ 0.001 and |log_2_Ratio| ≥ 1 (Figure 2, Appendix A). Among these significantly expressed genes, 2628 (45.1%) genes were upregulated and 3318 (54.9%) genes were downregulated in response to BD stress (Figure 2, Appendix A).

GO term analysis was performed; DEGs were classified into three categories, i.e., biological process, cellular component, and molecular function, and significantly enriched in 13 GO terms (Figure 3, Appendix A). This analysis revealed five biological processes, including microtubule-based process (GO:0007017), microtubule-based movement (GO:0007018), metabolic process (GO:0008152), carbohydrate metabolic process (GO:0005975), and photosynthesis (GO:0015979); three molecular functions, including microtubule motor activity (GO:0003777), transcription factor activity (GO:0003700), motor activity (GO:0003774); and five cellular components, including chromosomal parts (GO:0044427), chromosome (GO:0005694), chromatin (GO:0000785), nucleosomes (GO:0000786), and protein-DNA complexes (GO:0032993), were significantly enriched in the 13 GO terms (Figure 3, Appendix A).

Pathway analysis of the DEGs based on the KEGG database was performed to characterize the functional consequences caused by BD. The results certified that there are 11 pathways significantly enriched in response to BD, in which genes participating in metabolic pathways (26.65%), plant hormone signal transduction pathways (8.92%), and starch and sucrose metabolism (3.47%) were the top three pathways enriched under BD (Table 1). Other pathways, such as nitrogen metabolism; glycine, serine, and threonine metabolism; and ascorbate and aldarate metabolism were also enriched in response to BD (Table 1).

### 3.3. Candidate Genes Associated with Lignin, Cellulose, and Flavonol Synthesis Under BD

Lignin is biosynthesized from the phenylalanine pathway. Here, the candidate genes involved in lignin biosynthesis were identified as flow and regulation pathways. The results of global gene expression showed that numerous candidate genes were affected in the roots of “Nanlin 895” under BD conditions (Table 2 and Table 3 and Appendix A). Some were upregulated, and these genes included *PAL*, *CCR*, *CAD*, *COMT*, *F5H*, *PER*, and *LAC* (Table 2 and Appendix A). Additionally, a total of six identified candidate transcription factors (*MYB*(2), *TALE*(2), and *WRKY*(2); Appendix A) were upregulated. Some candidate genes also participated in hormone biosynthesis or signal transduction during cellular differentiation, including gibberellin receptors (GID1/2), gibberellin oxidases (gibberellin 20-oxidase/gibberellin 2-oxidase), sterol methyltransferase, chitinases (ethylene response factor), and some auxin-responsive proteins (Appendix A). These different expression patterns of transcription factors and signal transduction response genes are involved in the regulation of the dynamic accumulation of lignin during root development.

Flavonols were also synthesized from the phenylalanine pathway, and some of the major enzymes were identified. Interestingly, the CHS and F3’5’H genes identified were all downregulated (Figure 4; Table 2). This suggests that the deduced flavonol synthesis pathway may promote the upregulation of lignin. Cellulose-associated genes were also identified in this study, and five genes (hexokinase (3) and cellulose synthase (2)) were also downregulated (Table 2).

### 3.4. Validation of RNA-Seq Results Using qRT-PCR

To verify the results of the DEGs, nine DEGs were selected to analyze their expression levels using qRT-PCR in two libraries (Figure 5). The selected genes contained lignin monomer synthesis genes and transcription factors (Appendix A). The relative expression levels of CK and BD were comparable with those of the RNA-Seq data (Figure 4). Although the results of qRT-PCR were different from those of RNA-Seq, they all shared a similar direction of change between qRT-PCR and RNA-Seq (Figure 4). This comparison validated the results of RNA-Seq as accurate.

## 4. Discussion

### 4.1. BD Caused the Differential Expression of Lignin Biosynthetic Enzyme Genes

The phenylpropanoid pathway is responsible for the synthesis of a large variety of secondary metabolites, including phenol esters, coumarins, flavonoids, and lignin. Boron starvation has been found to significantly affect phenolic metabolism [26]. The expression levels of *CHSs*, *CHI*, and *F3’5’H*, which function as the rate-limiting enzymes in the flavonoid biosynthetic pathway (Figure 4, Table 2), were downregulated under BD, but this was not the case for the lignin monomer synthetic pathway. It can be hypothesized that the reduction of the flavonoid metabolic pathway compensated for the conversion of intermediate metabolites to the lignin biosynthetic pathway (Appendix A).

Phenylpropanoids, a key product in the biosynthesis of phenolic compounds, are catalyzed by phenylalanine ammonia-lyase (PAL). Previous reports suggest that *PtrPAL2,4,5* is expressed principally in xylem and root tips and mainly determines the production of lignin, which differs from the *PtrPAL1,3* function in producing the phenol metabolites [26]. In our investigation, the mRNA levels of *PtrPAL4* and *PtrPAL5* were upregulated more than 2 folds, which implies excess cinnamic acid promotes upregulation of the downstream monolignol biosynthesis genes and then contributes to the biosynthesis of monolignol in the early time period under BD [27]. The further mechanism still needs to be studied in the future.

Cinnamic acid is catalyzed by a series of enzymes and produces p-coumaroyl-CoA, which is then processed by cinnamoyl-CoA reductase (CCR) to coniferaldehyde, which in turn is converted to coniferyl alcohol by the action of CAD.

The downregulation of PtCAD1 results in a reduction of Klason lignin and an alteration of the proportion of monolignol. This result indicates that PtCAD1 is involved in lignin biosynthesis [28]. In this study, we found that BD induces the upregulation of Ptr CAD1, which may suggest that the accumulation of lignin is associated with its expression. The other *CAD* that showed upregulation may be induced by other stresses and, as described in a previous study [29], the function can be redundant, which requires additional verification. Although two CCRs were upregulated, we found that CCR2 [30], which was revealed as being involved in the biosynthesis of lignin, was not found, suggesting that the two CCRs may be involved in the lignin biosynthetic pathway.

*COMT* and *F5H* are both multifunctional enzymes. *F5H* functions in changing guaiacyl (G) monolignol to syringyl (S) monolignol and can utilize a large set of substrates, including ferulic acid, coniferaldehyde, and coniferyl alcohol. COMT is responsible for catalyzing O-methylation to a cluster of substrates, including caffeic acid, 5-hydroxy-coniferaldehyde, caffeoyl alcohol, 5-hydroxyl-feruloyl CoA, and 5-hydroxy-coniferyl alcohol [31]. Previously, the downregulation of *COMT* expression led to a decrease in lignin syringyl monomer content [32]. Consistently, the transcript levels of *F5H1* and *COMT1/2* were increased under BD, which implied that lignin accumulation was caused by increasing the intermediate product conversion of G and S lignin monomer biosynthesis. We also found that *4CL*, *C3H*, and *HCT*, the key enzymes of the lignin monomer biosynthetic precursor, decreased during BD, which implied the reduction of G and S lignin monomers and perhaps supported the increase of H:S [33].

Laccases and PER, both oxidases, are responsible for the last step of lignin biosynthesis. Disrupting *LAC4* and *LAC17* or overexpressing the *PRX* gene *AtPrx17* in Arabidopsis plants reduced or caused an increase in the lignin content, respectively [34,35]. The removal of hydrogen peroxide from the lignin-forming cell culture line of Norway spruce prevented or hindered the formation of lignin [36]. The increased expression levels of *LACs* and PERs, especially *LAC20* [37], *CWPO-C* [38], and potri.011g120300 and potri.009g042500 homologs with AtLAC17 and AtLAC4 may play key roles in lignin accumulation under BD.

### 4.2. Transcription Factors Involved in the Cell Wall Metabolic Process Under BD

BD can affect the expression of many transcription factors, such as *MYBs*, *WRKYs*, and *NAC*s [36]. Our data identified two *PtrMYBs* that shared a closer correlation with *AtMYB58* and *AtMYB63*, which are the second layer of the regulators of lignin synthesis. These two genes were upregulated in response to BD and may activate downstream monolignin biosynthetic enzyme genes. Some genes *(LAC4*, *LAC17*, *MYB46/83*, and *SND2)* related to SCW formation coexpressed with *BLH6*, and the overexpression of *BLH6* resulted in an accumulation of lignin deposition, and T-DNA mutants caused a decrease in lignin [33,39]. Our results are in accordance with these previously reported results. Our data demonstrated that two *TALE* genes (Potri004G159300 and Potri009G120800), which are homologous to *BLH6*, were upregulated in response to BD. Lignin biosynthesis is controlled by a regulatory cascade of upstream transcription factors controlling the formation of secondary walls by activating several other transcription factors, including *MYB43*, *MYB46*, *MYB58*, *MYB63*, *MYB85*, *MYB103*, *SND3*, *KNAT7*, *BLH6*, and *WRKY12* [40,41,42]. Three *WRKY* genes (Potri001G328000, Potri002G193000, and Potri008G103300) (Table 2) shared closer correlations with *WRKY12*, suggesting that these genes play critical roles in regulating the upregulation of monolignin biosynthetic enzyme genes, but the specific roles still need to be studied.

### 4.3. Hormones Participate in the Signal Transduction of Cell Wall Formation Under BD

Plant hormones have been inextricably linked with BD and cell wall formation [43,44,45]. In vascular plants, lignification occurs only in certain types of cells, such as tracheary elements and fiber cells. To adjust themselves to the changing environment and thicken the cell wall, meristematic cells must receive signals that initiate the cell differentiating and cell wall thickening programs. Changes in phytohormone levels are linked with the synthetic and metabolic pathway genes [46]. Here, we have demonstrated that some *GA20OXs* and *GIDs* function as synthases and that the receptors of GA are upregulated; *GA2OXs* catalyze the deactivation of bioactive GA and are downregulated. Consistent with this, the overexpression of *AtGA20ox* in transgenic tobacco plants showed a higher content of lignin in the *AtGA20ox* plants compared to wild-type controls [47,48], implying that the increased levels of GA may cause the accumulation of lignin. In contrast, some results demonstrated that spraying winter wheat with exogenous GA_3_ significantly decreased lignin accumulation compared to the control, suggesting that GA interacts with other hormones to function during secondary cell wall formation under boron deficiency [49].

Auxin was considered to take part in the cell expansion and the biosynthesis of the cell wall by repression of the *KNOX* gene [50,51]. The expression levels of the genes encoding auxin-responsive proteins were found to be downregulated, which means that auxin levels decreased under BD and agreed with the results of Bairu et al. [52]. The mutant *iar4* leading to the functional loss of auxin could result in the accumulation of lignin [51]. Ban et al. [53] found that the application of exogenous 2,4-D in high concentrations can inhibit the expression of *PAL*, which may support the hypothesis that low auxin can oppose the inhibition of *PAL* expression in different directions. Some other reports showed that BD reduces the elongation of cells but not the division, and three receptor genes of cytokinin were found to be upregulated in the DEGs, which agrees with previous results [54].

Brassinosteroids (BRs) can also protect plants from environmental stresses [55]. Sterol synthetic mutants of sterol methyltransferase 1 showed a deficiency of BRs, cellulose, and the ectopic deposit of lignin but not other cell wall components, such as the content of neutral sugars and pectin [56], suggesting that the downregulation of two sterol methyltransferases decreased the BR content. The reduction of the cellulose content further inhibited cell elongation, which not only compensated for the lignin content but also increased the total content of the cell wall by increasing the superficial area of the cell wall [56].

Jasmonic acid (JA) has been shown to be a significant regulator of plant responses to environmental stresses [46,57]. A reduction in cellulose biosynthesis induces the production of the phytohormones JA and ethylene, resulting in ectopic lignin production [57]. *coi1* (impaired in *JA-Ile* perception) seedlings all exhibit enhanced lignin deposition [57]. With the downregulation of cellulose synthase and upregulation of JA-related genes, it can be considered that cellulose reduction compensates for the lignin content, and to maintain cell proliferation, the increased JA promotes the irregular distribution of lignin under BD.

Similar to the results above, the chitinase-like gene was reported to play a critical role in the biosynthesis of the cell wall, which is regulated by ethylene [58]. Zhong et al. [58] showed that the chitinase-like mutant manifested a reduction of cellulose and the deposition of lignin. Interestingly, we found that half (3 out of 6) of the chitinase genes were upregulated and may take part in other metabolic pathways. The specific functions of these chitinases still merit additional study.

The findings of this study indicated that plant hormones, such as GA, BR, ethylene, auxin, and cytokinin, play important roles in the modulation of lignin biosynthesis under BD. However, their roles in the regulation of lignin synthesis under BD still need to be studied at a deeper level.

## 5. Conclusions

In this study, RNA-Seq of the roots of “Nanlin 895” was performed and analyzed. Through the high-throughput sequencing technique, we identified some genes involved in phytohormone (including auxin, BR, GA, ETH, JA, and CTKs) metabolism and signaling, cell elongation (XTH), cell division (CYCD), lignin biosynthesis (including PAL, COMT, F5H, CCR, CAD, LAC and PER), cellulose (CesAs, hexokinase), and flavonol synthesis (CHS, CHI, F3’5’H), as well as transcription factors such as *WRKY*, *TALE*, and *MYB*. These results provide an important theoretical basis and reference data in plant for further research on the mechanism of lignin accumulation under BD.

## Figures and Tables

**Figure 1 biomolecules-09-00156-f001:**
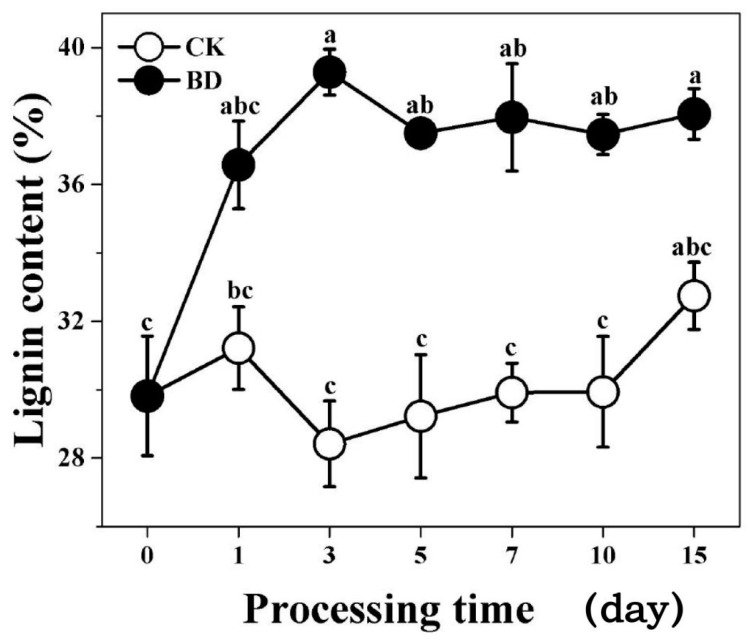
Lignin content in the “Nanlin895” roots in different treatment periods. The hollow circle and solid circle lines represent the boron deficiency (BD) and control (CK) treatments, respectively. Data are expressed as the mean ± SE (*n* = 3). ^a,b,c^ Different letters above the error bars indicate significant differences at the 0.05 level.

**Figure 2 biomolecules-09-00156-f002:**
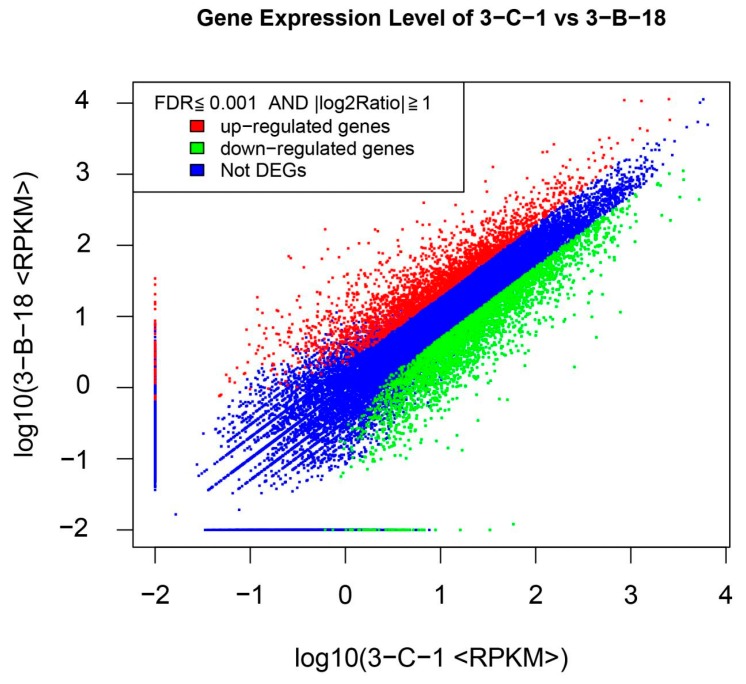
Number of significantly differentially expressed genes in “Nanlin895” roots exposed to boron deficiency.

**Figure 3 biomolecules-09-00156-f003:**
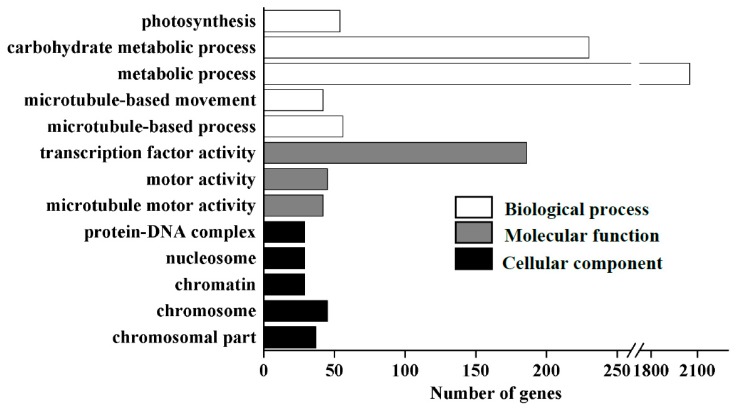
The significantly enriched GO terms for differentially expressed genes in “Nanlin895” roots exposed to boron deficiency. The x-axis and y-axis indicate the names of clusters and gene number assigned to the cluster, respectively.

**Figure 4 biomolecules-09-00156-f004:**
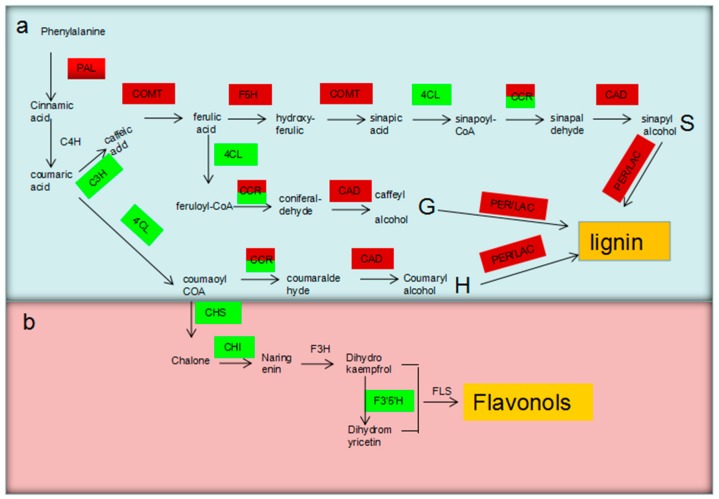
Simplified biosynthetic pathway of lignin and flavonols. (**a**) The lignin biosynthetic pathway; (**b**) the biosynthetic pathway of flavonols; upregulated and downregulated genes are denoted with red and green backgrounds, respectively, while nondifferentially expressed genes are denoted with no background.

**Figure 5 biomolecules-09-00156-f005:**
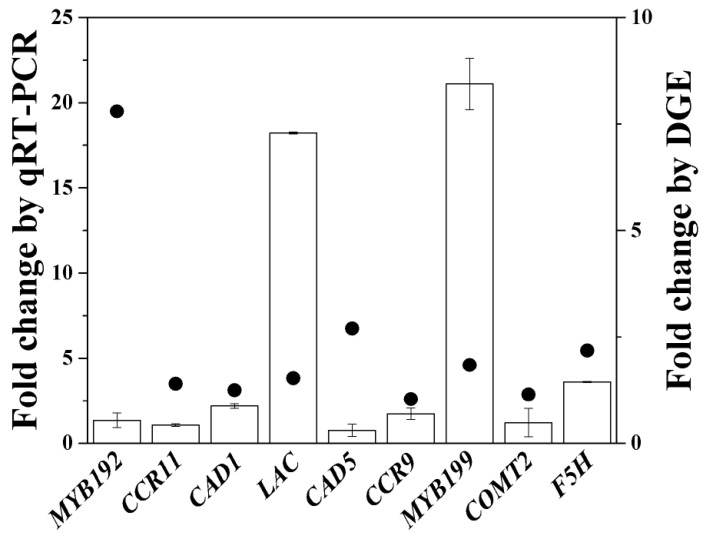
qRT-PCR confirmation of differentially expressed genes from digital gene expression analysis. ● Solid circles and histograms represent the results obtained from RNA-Seq and qRT-PCR, respectively. Bars indicate the mean ± SE (*n* = 3). Detailed information on the selected genes is available in Appendix A.

**Table 1 biomolecules-09-00156-t001:** Significantly enriched KEGG pathways of significantly differentially expressed genes in roots of “Nanlin895” exposed to boron deficiency.

Pathway ID	Pathway	DEGs with Pathway Annotation	All Genes with Pathway Annotation	*q*-Value
ko01100	Metabolic Pathways	983/3688 (26.65%)	5504/23453 (23.47%)	6.13 × 10^5^
ko00196	Photosynthesis—Antenna Proteins	15/3688 (0.41%)	28/23453 (0.12%)	2.60 × 10^−4^
ko04075	Plant Hormone Signal Transduction	329/3688 (8.92%)	1742/23453 (7.43%)	5.38 × 10^−3^
ko04712	Circadian Rhythm—Plant	57/3688 (1.55%)	232/23453 (0.99%)	9.41 × 10^−3^
ko00053	Ascorbate and Aldarate Metabolism	43/3688 (1.17%)	165/23453 (0.7%)	1.05 × 10^−2^
ko00260	Glycine, Serine, and Threonine Metabolism	33/3688 (0.89%)	118/23453 (0.5%)	1.05 × 10^−2^
ko00910	Nitrogen Metabolism	25/3688 (0.68%)	84/23453 (0.36%)	1.55 × 10^−2^
ko00500	Starch and Sucrose Metabolism	128/3688 (3.47%)	632/23453 (2.69%)	1.96 × 10^−2^
ko00051	Fructose and Mannose Metabolism	32/3688 (0.87%)	120/23453 (0.51%)	2.02 × 10^−2^
ko00072	Synthesis and Degradation of Ketone Bodies	7/3688 (0.19%)	14/23453 (0.06%)	3.64 × 10^−2^
ko00052	Galactose Metabolism	32/3688 (0.87%)	126/23453 (0.54%)	3.82 × 10^−2^

**Table 2 biomolecules-09-00156-t002:** Selected significantly differentially expressed genes related to phenylpropanoid biosynthesis and cellulose biosynthesis in roots of “Nanlin895” exposed to boron deficiency.

Gene ID	Symbol	Annotation	Log_2_ Ratio	Homology
***PAL***				
Potri.010G224100	*PtrPAL4*	phenylalanine ammonia-lyase	1.22	AT2G37040
Potri.010G224200	*PtrPAL5*	phenylalanine ammonia-lyase	1.58	AT2G37040
***CCR***				
Potri.009G076300	*PtrCCR9*	cinnamoyl-CoA reductase	1.04	AT5G58490
potri.017g110500			−1.45	AT5G14700
potri.013g079500			−2.89	AT2G23910
Potri.002G004500	*PtrCCR11*	cinnamoyl-CoA reductase	1.40	AT2G33590
***CAD***				
Potri.001G256400	*PtrCAD*	cinnamoyl-CoA reductase	1.24	AT5G19440
Potri.006G024300	*PtrCAD16*	cinnamyl-alcohol dehydrogenase	3.33	AT1G72680
Potri.009G095800	*PtrCAD1*	cinnamyl-alcohol dehydrogenase	1.25	AT4G34230
Potri.009G063100	*PtrCAD3*	cinnamyl-alcohol dehydrogenase	1.95	AT4G37990
Potri.009G062800	*PtrCAD5*	cinnamyl-alcohol dehydrogenase	2.70	AT4G37990
Potri.001G268600	*PtrCAD7*		−1.38	AT4G37990
Potri.001G300000		cinnamyl-alcohol dehydrogenase	2.71	AT4G34230
***F5H***				
Potri.005G117500	*PtrCAld5H1*	ferulate-5-hydroxylase	2.18	AT4G36220
**COMT**				
Potri.015G003100	*PtrCOMT1*	caffeic acid 3-O-methyltransferase	1.34	AT5G54160
Potri.012G006400	*PtrCOMT2*	caffeic acid 3-O-methyltransferase	1.15	AT5G54160
Potri.014G106600	*PtrCOMT3*	caffeic acid 3-O-methyltransferase	−4.20	AT5G54160
Potri.011G059500	*PtrCOMT8*	caffeic acid 3-O-methyltransferase	2.10	AT4G35160
Potri.004G050400	*PtrCOMT9*	caffeic acid 3-O-methyltransferase	4.10	AT4G35160
Potri.001G451100	*PtrCOMT25*	caffeic acid 3-O-methyltransferase	2.43	AT5G54160
***PER***				
Potri.006G107000		Peroxidase	3.44	AT5G05340
Potri.017G064100		Peroxidase	2.87	AT5G67400
Potri.018G136900		Peroxidase	2.29	AT4G33420
Potri.009G106400		Peroxidase	2.19	AT1G49570
Potri.016G132800		Peroxidase	2.02	AT1G14550
Potri.T045500		Peroxidase	1.88	AT4G33420
potri.016g125000	*CPWPOC*	Peroxidase	1.12	AT2G41480
***4CL***				
potri.010g057000		4-coumarate-CoA ligase	−1.69	AT5G63380
potri.003g099700		4-coumarate-CoA ligase	−2.76	AT4G19010
potri.019g049500		4-coumarate-CoA ligase	−1.21	AT1G65060
potri.t071600		4-coumarate-CoA ligase	−1.04	AT1G65060
potri.002g012800		4-coumarate-CoA ligase	−1.99	AT1G20510
***C3H***				
potri.016g031100			−2.92	AT2G40890
Potri.016G031000		coumaroylquinate (coumaroylshikimate) 3’-monooxygenase	−2.99	AT2G40890
potri.019g130700	C4H 2		−1.34	AT2G30490
potri.018g146100	C4H3		−2.55	AT2G30490
***HCT***				
Potri.018G105500	*PtrHCT2*	shikimate O-hydroxycinnamoyl transferase	−1.19	AT5G48930
***LAC***				
potri.011g120300			1.45	AT5G60020
potri.009g042500			1.24	AT2G38080
Potri.010G183500			1.53	AT2G40370
potri.009g034500	*Pt-LAC20*		1.40	AT2G29130
**Cellulose Synthesis**				
Potri.016G054900	*Pt-CESA4.2*	similar to cellulose synthase	−1.21	AT5G05170
Potri.006G052600	*Pt-CESA4.1*			
potri.018g088300		Hexokinase	−1.38	AT2G26310
potri.001g190400		Hexokinase	−1.50	AT4G29130
potri.009g050000		Hexokinase	−1.16	AT1G50460
**Flavonols Synthase**				
Potri.003G176900		chalcone synthase	−1.05	AT5G13930
Potri.003G176800		chalcone synthase	−1.03	AT5G13930
Potri.003G176700		chalcone synthase	−1.03	AT5G13930
potri.006g219600		chalcone isomerase	−1.09	AT2G26310
potri.009g069100		flavonoid 3′,5′-hydroxylase	−1.97	AT5G07990

**Table 3 biomolecules-09-00156-t003:** Significantly differentially expressed transcription factors and phytohormone related genes in roots of ‘Nanlin895’ exposed to boron deficiency.

Gene ID	Symbol	Annotation	Log_2_ Ratio	FDR
***WRKY***				
Potri.002G193000	*Pt-WRKY48.2*	WRKY transcription factor 33	2.00	8.54 × 10^−73^
Potri.008G103300		WRKY transcription factor 33	1.17	2.87 × 10^−9^
***TALE***				
Potri.004G159300			1.58	4.66 × 10^−61^
Potri.009G120800			2.22	2.06 × 10^−106^
***MYB***				
Potri.007G067600	*MYB192*	myb proto-oncogene protein	7.80	2.81 × 10^−5^
Potri.012G127700	*MYB199*		1.80	1.61 × 10^−55^
**Signal Transduction Pathway**				
**Gibberellic Acid**				
Potri.008G180500		gibberellin receptor GID1	2.21	1.05 × 10^−44^
Potri.013G028700		gibberellin receptor GID1	1.25	1.71 × 10^−21^
Potri.014G135900		gibberellin receptor GID1	1.17	3.93 × 10^−30^
Potri.016G065000		gibberellin receptor GID1	1.02	1.94 × 10^−21^
Potri.005G208200		F-box protein GID2	2.15	1.74 × 10^−12^
Potri.014G022100		F-box protein GID2	1.40	5.46 × 10^−61^
Potri.002G122300		F-box protein GID2	1.36	2.29 × 10^−30^
Potri.010G060800		F-box protein GID2	1.03	2.59 × 10^−5^
Potri.007G103800	*GA20ox5*	gibberellin 20-oxidase	3.91	3.92 × 10^−10^
Potri.015G134600	*GA20ox8*	gibberellin 20-oxidase	1.88	2.87 × 10^−9^
Potri.001G176500	*2OGox4*	gibberellin 20-oxidase	1.66	1.23 × 10^−51^
Potri.009G107600	*2OGox7*	naringenin 3-dioxygenase	1.07	3.84 × 10^−19^
**Bressionsteroid**				
Potri.001G263700		sterol 24-C-methyltransferase	−2.06	3.60 × 10^−26^
Potri.005G245800		sterol 25-C-methyltransferase	−1.81	1.08 × 10^−223^
**Ethylene**				
Potri.009G142300		Chitinase	−3.22	1.26 × 10^−25^
Potri.002G186500		Chitinase	−2.14	2.99 × 10^−15^
Potri.009G141800		Chitinase	−3.39	1.30 × 10^−4^
Potri.015G024200		Chitinase	1.01	3.72 × 10^−73^
Potri.004G182000		Chitinase	1.77	2.91 × 10^−43^
Potri.T175200		Chitinase	3.94	2.12 × 10^−10^
**Auxin**				
Potri.005G218200	*Pt-AUX22.4*	auxin-responsive protein IAA	−4.32	4.21 × 10^−5^
Potri.003G056900	*Pt-AUX22.3*	auxin-responsive protein IAA	−3.28	9.00 × 10^−8^
Potri.010G078300	*Pt-IAA14.2*	auxin-responsive protein IAA	−2.55	1.91 × 10^−71^
Potri.008G161200	*Pt-IAA14.1*	auxin-responsive protein IAA	−2.50	1.40 × 10^−293^
Potri.001G177500		UDP-N-acetylglucosamine	−2.22	2.33 × 10^−11^
Potri.006G166900		auxin-responsive protein IAA	−2.22	3.94 × 10^−4^
Potri.002G045000	*Pt-AUX22.5*	auxin-responsive protein IAA	−1.56	3.46 × 10^−5^
Potri.006G161400		auxin-responsive protein IAA	−1.02	1.92 × 10^−5^
Potri.005G174000	*PtrAUX7*	auxin-responsive protein IAA	−2.04	3.29 × 10^−57^
Potri.002G087000	*PtrAUX8*	auxin-responsive protein IAA	−1.20	3.49 × 10^−13^
Potri.016G113600	*PtrAUX2 Pt-AUX1.1*	auxin-responsive protein IAA	−1.05	2.86 × 10^−61^
Potri.004G172800	*PtrAUX5*	auxin-responsive protein IAA	−1.02	7.08 × 10^−6^
Potri.018G139400	*PIN9 Pt-PIN2.4*	auxin efflux carrier family	−3.09	6.01 × 10^−64^
Potri.005G187500	*PIN4 Pt-PIN6.2*	auxin efflux carrier family	−2.63	7.05 × 10^−9^
Potri.016G035300	*PIN2 Pt-PIN2.3*	auxin efflux carrier family	−1.25	1.45 × 10^−9^
Potri.012G047200	*PIN7 Pt-PIN1.2*	auxin efflux carrier family	−1.10	8.28 × 10^−9^
Potri.009G132100	*PtrAUX6 Pt-LAX5.1*	auxin influx carrier (AUX1 LAX family)	−2.79	2.80 × 10^−38^
Potri.011G042400	*TIR*	transport inhibitor response 1	−1.07	1.16 × 10^−5^
**CTKS**				
Potri.007G056400	*Pt-ATHK1.2*	similar to histidine kinase 1	1.87	9.39 × 10^−7^
Potri.003G171000	*Pt-AHK3.2 PHK4*	similar to histidine kinase receptor	1.71	4.79 × 10^−21^
Potri.005G111700	*Pt-ATHK1.3*	similar to histidine kinase 1	1.36	1.00 × 10^−35^
**JA**				
Potri.010G192900		coronatine-insensitive protein 1	1.19	6.47 × 10^−55^

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
