# Peer review of "Global Transcriptomic Profile Analysis of Genes Involved in Lignin Biosynthesis and Accumulation Induced by Boron Deficiency in Poplar Roots"

_biomolecules, 2019, doi:10.3390/biom9040156_

Reviewer 1 Report

The authors have addressed several criticisms raised by the reviewers, the revised manuscript has been improved, in particular the discussion of the results is now adequate.

There are still some points that need attention:

 Lines 70-72

As observed by Rewiever #2, the use of root tips from only 3 plants (3 biological replicates) for RNA-seq analysis seems limiting. Furthermore, it is not clear why the authors chose the roots with the highest lignin content since the lignin content measured in the root tips after 3 days of B deprivation is much higher in B deprived than in B sufficient plants. It could be biased. The authors should comment this point.

“For the saplings, we used three plants to determine the lignin content, three for RNA-Seq, three for qRT-PCR, and  others as additional materials. Then, the root tips with the highest lignin content were selected to  extract RNA.”

 Lines 343-345

“Through the high-throughput sequencing technique, we identified some genes involved in transporters (NIP, 344 boron transporter1-related).”

In the conclusions,  the authors mentioned genes coding for transporters including a Boron transporter, but these genes are not discussed in the results and it is not clear how their expression is modified under Boron deprivation.

 The following sentences are not clear and should be rephrased:

 Lines 60-63 

 “In early July, two-months-old, approximately 30-40 cm height from tissue culture saplings with no pests, good growth and uniform size were chosen 61 as the materials, precultured in nutrient ..“

Lines 348-350

“These gene findings can provide an important evidence for further research on the expression , regulation  mechanism of lignin accumulation under BD.”

 Author Response

Dear reviewer:

 We would like to thank you for the thoughtful and thorough comments and suggestions. They have been very helpful in resulting in a much improved manuscript. Detailed responses to specific comments are listed below.

 Response to reviewer:

Comment: As observed by Rewiever #2, the use of root tips from only 3 plants (3 biological replicates) for RNA-seq analysis seems limiting.

Response: We agree with you the view ,and we will pay more attention to the problem and correct it in the future researches.

  Comment: Furthermore, it is not clear why the authors chose the roots with the highest lignin content since the lignin content measured in the root tips after 3 days of B deprivation is much higher in B deprived than in B sufficient plants. It could be biased. The authors should comment this point.

(Line 71-73,“For the saplings, we used three plants to determine the lignin content, three for RNA-Seq, three for qRT-PCR, and  others as additional materials. Then, the root tips with the highest lignin content were selected to  extract RNA.”)

 Response: We are very sorry for the description above could not express exactly our process. In fact ,we hypothesis the content of lignin treatment group( boron 0mg) is significantly higher than that of Control( 0.25 mg L-1) and shows a certain trend with the prolonging of time. To find a apporiate stage for RNA-seq, we assumed sample with the highest lignin content could respresent the characaterzation of treatment group and Control group Thus the  samples with highest lignin content were projected to used for RNA-seq. After we obtain the result(Figure 1), we found the lignin content is signinficantly different between Control group and Treanment group from the 3rd day and no signinficantly different within the group. It means that 3rd,5th,7th,10th all could be used for RNA-seq, and the random selection method was used to select the samples for RNA-seq. In a word, we selected the 3 day to perform the RNA-seq is result of random selection according to the lignin content.

 Comment: In the conclusions,  the authors mentioned genes coding for transporters including a Boron transporter, but these genes are not discussed in the results and it is not clear how their expression is modified under Boron deprivation.

“Through the high-throughput sequencing technique, we identified some genes involved in transporters (NIP, 344 boron transporter1-related).

Response: In this study, we focused on the pathway and regulatory mechanism of lignin accumulation under boron deficiency ,havent pay much attention to the boron transporter genes, although we found two NIPs were downregulated and considering the eassay structure and completeness, we have decided to do not discuss the expression and function of the boron transporter genes and deleted the description section of boron transporter genes in conclusions section.

Comment:“In early July, two-months-old, approximately 30-40 cm height from tissue culture saplings with no pests, good growth and uniform size were  chosen 61 as the materials, precultured in nutrient(line 59-62) .

Response: we have altered the description as  In early July, about two-months-old, approximately 30-40 cm height saplings with no pests, good growth and uniform size were taken out from the pots, cleaned up and precultured in nutrient.(line 59-62)

Comment:“These gene findings can provide an important evidence for further research on the expression , regulation  mechanism of lignin accumulation under BD.”

Response: We have rewrite the sentence as  These results provide an important theoretical basis and reference data in plant for further research on the mechanism of lignin accumulation under BD(line 355-357).”

 Reviewer 2 Report

Review of the article "Global Transcriptomic Profile Analysis of Genes Involved in Lignin Biosynthesis and Accumulation Induced by Boron Deficiency in Poplar Roots".

The article topic is up-to-date, and the presented results are actual and valuable. The aim of the work is clear and important.

In my opinion, this article can be published in the Biomolecules journal.

Author Response

Dear reviewer:

Thank you for your acceptance!

Reviewer 3 Report

In this manuscript, the authors did some research on the accumulation mechanism of lignin in poplar. Under the condition of boron deficiency, the content of lignin significantly increased after 3 days incubation. Using RNA-Seq, they found many genes were regulated. Among them, some genes level related to lignin monomer biosynthetic pathway were upregulated. But about hormone signal regulating genes, some were upregulated and some were downregulated. These finds may be helpful to analyze lignin formation. However, there are several problems with this research that may affect the results.

1. Line 62

Hoagland's nutrient solution contains 0.5 mg/L boron. Is this solution used in this manuscript commercial or self-prepared? How did you remove the contaminant boron from other components?

2. Line 74

The authors gave two methods to determine lignin content. What's the difference when they are applied to detect the known lignin? In Fig. 1, the lignin content was determined only by ultraviolet spectrophotometry. So what purpose is Klason's method?

3. Line 98-107

There are several methods to make library for RNA-seq. Each method has its own bias. The authors should describe the detailed steps about RNA-seq library preparation in the main text or as the supplement, including reagents, oligos, enzymes used. Otherwise, any results of RNA-seq will not be trusted.

4. Line 163

Which day's sample was used to extract RNA for sequencing? Are samples of CK and BD from the same day or different day?

5. Line 189

The authors should explain the reasons why the analysis of downregulating genes was not included, although some were mentioned in the later.

6. Line 225

According to the results of qRT-PCR, 4 results showed a great difference to RNA-seq although they were the same direction. Which result should be trusted?

7. Line 249

The authors should explain how the upregulation of mRNA (PtrPAL4 and PtrPAL5) would affect the biosynthesis of monolignol?

Author Response

Dear reviewer:

 We would like to thank you for the thoughtful and thorough comments and suggestions. They have been very helpful in resulting in a much improved manuscript. Detailed responses to specific comments are listed below.

 Response to reviewer

Comment: Hoagland's nutrient solution contains 0.5 mg/L boron. Is this solution used in this manuscript commercial or self-prepared? How did you remove the contaminant boron from other components?

Response: In this study the Hoagland's nutrient solution were self-prepared and the components were devided into five parts. Boric acid was  separate from other components. The work concentration of boric acid added according to the experimental requirements.

Conment: The authors gave two methods to determine lignin content. What's the difference when they are applied to detect the known lignin? In Fig. 1, the lignin content was determined only by ultraviolet spectrophotometry. So what purpose is Klason's method?

Response: 1)Compared with Klasons method ,ultraviolet spectrophotometry method is rapid, accurate and suitable for the determination of lignin content in ginseng and other precious Chinese medicinal materials. Klason method only determined the acid insoluble lignin and ignored the soluble lignin of plants;

2)For the calculation of standard light absorption(Astandard), Klason methods [19]were used to estimate the lignin content of the untreated samples [19].

In brief, root powder samples (w1) were placed into a mortar with 20 mL of 72% H2SO4 and incubated at room temperature for 4 h. Then, the samples were mixed with ca. 765 mL of deionized water and incubated in boiling water for 2 h. The mixture was filtered with a sand core funnel (w2), and the funnel with lignin was dried to a constant weight (w3). The calculation of the lignin content (Klignin) follows the format that lignin %=(w3-w2)/w1×100%. Absorptivity was calculated as Absorptivity=Abs×liters/Wsamples×klignin, Astandard were calculated as average number of different Absorptivity of biological replicates. For the lignin content of different stages of root samples, ultraviolet spectrophotometry was carried out [20]. Briefly, ca. 6 mg of root powder samples were mixed with 5 mL of 25% (w/w) acetyl bromide acetic acid and 0.2 mL of perchloric acid, sequentially. The tubes were sealed and incubated at 70℃ for 30 min, and then the mixture was mixed with 10 mL of 2 M NaOH and diluted with glacial acetic acid to 100 mL. The lignin contents in the samples were determined spectrophotometrically at 254 nm (the maximum absorption wavelength was obtained by scanning the absorption peak using ultraviolet-visible spectroscopy from 200 - 600 nm). The calculation of the lignin contents was performed via the formula lignin%=Abs×liters×100%/Wsamples×Astandard.

 Comment: There are several methods to make library for RNA-seq. Each method has its own bias. The authors should describe the detailed steps about RNA-seq library preparation in the main text or as the supplement, including reagents, oligos, enzymes used. Otherwise, any results of RNA-seq will not be trusted.

Response: We added more detail information of reagents about the RNA-seq library construction , include product name, product code, manufacturer, operational procedure 

Comment: Which day's sample was used to extract RNA for sequencing? Are samples of CK and BD from the same day or different day?

Response: The samples at stage 3 day were used to extract RNA for sequencing. The samples of CK and BD from were from the same day.

Comment: The authors should explain the reasons why the analysis of downregulating genes was not included, although some were mentioned in the later.

Response: We focus on boron deficiency inducing the increase of lignin accumulation. Although there are many researches report their achievemnt on lignin biosynthesis and regulation, there are still very few report about the negative regulation genes were research,so its difficult to analysize the down regulation genes.

Comment: According to the results of qRT-PCR, 4 results showed a great difference to RNA-seq although they were the same direction. Which result should be trusted?

 Response: In terms of samples and RNA quality,the rusults of RNA-seq are more credible, because of the qR-PCR samples were stored longer time,which may have impact onthe result . But as they present the same direction this may indicate that these genes responsed during the boron deficiency, in this regard these results all credible.

Comment: The authors should explain how the upregulation of mRNA (PtrPAL4 and PtrPAL5) would affect the biosynthesis of monolignol?

Response: We have rewrite and explain the upregulation of mRNA (PtrPAL4 and PtrPAL5)  affect the biosynthesis of monolignol.(line 256-260 ).

Round  2

Reviewer 3 Report

1) Because only unltrviolet spectrophotometry method was used, the description of Klason’s method should be deleted.

2)If the authors would not like to show detailed steps of RNA-seq libraries construction, the alternative is that a known method is cited or to tell us which company did that. The authors should understand that the detailed steps are so helpful and important for readers to judge whether the result is useful or not.

Author Response

Comment: Because only unltrviolet spectrophotometry method was used, the description of Klason’s method should be deleted.

Response: Thank you for your suggesstion. We have deleted the the description of Klason’s method .

Comment: If the authors would not like to show detailed steps of RNA-seq libraries construction, the alternative is that a known method is cited or to tell us which company did that. The authors should understand that the detailed steps are so helpful and important for readers to judge whether the result is useful or not.

Response: Thank you for your suggesstion. We have added the information of the company for RNA-Seq library construction and high throughput sequencing.(line 98-99)

This manuscript is a resubmission of an earlier submission. The following is a list of the peer review reports and author responses from that submission.

Round  1

Reviewer 1 Report

The manuscript entitled ”Global Transcriptomic Profiles Reveal Lignin Accumulation Induced by Boron Deficiency in Poplar Root” by Su W. and collaborators describes the transcriptomic changes associated to increased lignin content in poplar roots. The data are only partially novel as similar results have been obtained in other plant species. In addition, the manuscript presents methodological problems and overinterpretation of the results. The English is poor, the text contains errors and inconsistencies.

The authors analysed the root transcriptome after 3 days of boron deprivation assuming that this treatment induces boron deficiency. This assumption should be supported by the measurement of boron in the plant tissues. In addition, the age of the saplings is not clearly stated, I think that the induction of boron deficiency can depend on the age of the plant too. The authors chose the time for transcriptome analysis on the base of the root lignin content (at 3 days they measured the highest content of lignin in boron-deprived roots). However, I have some concerns about the measurements of lignin content. There are several methods to analyse the lignin content in plant tissues, also depending on the starting plant material. The authors used two different methods which implies different ways to calculate the lignin content. In my opinion, they should present the results of both methods for comparison. The methods for measuring lignin content are not described in detail and need references. For instance, in the case of the acetyl bromide method, how the standard curve was prepared? The content of lignin is calculated as percentage of root fresh weight, while is usually calculated as percentage of the dry weight of cell walls.

The description of the qRT-PCR methods is too generic qRT-PCR was performed according to the ZHUANGMEN manufacturer specifications.“

The transcriptome analysis identified 5956 DEG, only a small subset of these DEG are commented in the text (genes related to lignin synthesis, some transcription factors and genes related to hormone signalling and metabolism). The authors should justify this choice, as other categories of genes seem well represented among DEG.

The model reported in Fig.5 is not fully substantiated by the data obtained, it appears too speculative. Figure 2 is useless.

 Reviewer 2 Report

. As stated, I consider that the manuscript is not acceptable:

- The quality of the written English must be improved throughout the manuscript.

- Authors used a very poor plan of experimentation. If I understand well, they used root tips from only 3 plants for lignin contents, root tips from a pool of 3 plants for RNA-seq (only one sequenced sample) and root tips from 3 plants for qRT-PCR (the same 3 plants as used for RNA-seq??).  In M&M section, they claimed that they used 12 independent biological replicates (in fact, 12 plants), but I cannot see in the manuscript how they used these 12 plants?

-Authors are too affirmative when they claimed that they have "a deep understanding of lignin bio-synthesis pathway". In this paper, authors only identified gene deregulation; there is no functional analyses to validate the data. All the comments can be only speculative. The title must be modified: transcriptomic profiles did not reveal lignin accumulation. Figure 5 must be removed or the legend must be completely reformulated. The discussion and conclusions sections must be rewritten in this way.